# A Novel Signal Design and Performance Analysis in NavCom Based on LEO Constellation

**DOI:** 10.3390/s21248235

**Published:** 2021-12-09

**Authors:** Jing Ji, Yuting Liu, Wei Chen, Di Wu, Hongyang Lu, Jiantong Zhang

**Affiliations:** 1School of Information Engineering, Wuhan University of Technology, Wuhan 430070, China; jijingisme@whut.edu.cn (J.J.); luhongyang@whut.edu.cn (H.L.); 2School of Automation, Wuhan University of Technology, Wuhan 430070, China; liuyuting@whut.edu.cn; 3Key Laboratory of Environment Change and Resources Use in Beibu Gulf, Nanning Normal University, Nanning 530001, China; wudi324243@163.com; 4Department of Navigation, China Transport Telecommunications and Information Center, Beijing 100010, China; mygis2014@163.com

**Keywords:** signal design, LEOs, NavCom, BDS, 5G/B5G, CE-OFDM-PM

## Abstract

The mega-launch of low Earth orbit satellites (LEOs) represents a critical opportunity to integrate navigation and communication (NavCom), but first, challenges related to signal design must be overcome. This article proposes a novel signal scheme named CE-OFDM-PM. Via research on the in-band or adjacent band, it was found that the proposed signal scheme was suitable for S-band and had a wide normalized power spectrum density (PSD), high peak-to-side lobe ratio (PSR), and multiple peaks in autocorrelation. In an analysis of the simulation performance evaluation in navigation and communication, it is found that the proposed signal scheme has the potential for high accuracy, a code tracking accuracy of up to 0.85 m, a small mutual influence between the proposed signal scheme and other schemes, excellent anti-interference properties, and a better performance at both short and long distances in terms of its anti-multipath capability. Furthermore, the proposed signal scheme shows the ability to communicate between satellites and the ground and is outstanding in terms of its bit error rate (BER), CNR, and energy per bit to noise power spectral density ratio (Eb/N0). From the technical, theoretical, and application perspectives, our proposed signal scheme has potential as an alternative scheme in future BDS, PNTs, and even 5G/B5G.

## 1. Introduction

The global navigation satellite system (GNSS), as a national critical infrastructure (NCI), offers position, velocity, and time (PVT) services to various users around the world. Recently, the phase 3 BeiDou navigation satellite system (BDS) has also assisted in the launch of the 31st satellite. In this era of global networking [1], it is the third GNSS system to provide global spatio-temporal services after GPS and GLONASS. Due to the vulnerability of the GNSS system itself, the service cannot meet various requirements, especially requirements related to accuracy, availability, integrity, and anti-jamming indicators, directly leading to huge restrictions being placed on satellite navigation applications.

To meet the diversification requirements for positioning, navigation, and timing (PNT) capabilities, communication and remoting technologies can be used to extend signal coverage, called PNTRC. The concept of PNTRC was proposed to integrate positioning, navigation, timing, remoting, and communication services into one concept [2]. It corresponds to integration sensing and communication (ISAC) in B5G/6G and has received positive responses from Chinese scholars such as Prof. Jingnan Liu, Prof. Yuanxi Yang, and Prof. Deren Li [3,4,5]. The concept of integrated navigation and communication (NavCom) was first proposed in a U.S. report in 1977 [6]. Since then, the concept of NavCom has been adopted for the spatio-temporal grid and been widely mentioned in various fields. The concept of Aided GPS (A-GPS) was first proposed in the U.S. and originated from the application of inertial navigation and the Loran-C system in the maritime industry [7,8]. However, the Boeing company led the combined Iridium with the GPS (iGPS) program and building of an LEO/MEO satellite hybrid system that combined communication satellite assets, including Iridium, to provide users with enhanced services and innovations regarding the integrity of iGPS [9]. In 2013, Germany proposed the L-band-based LDACS1 solution, based on the OFDM broadband system combined with the L-band broadband multi-carrier communication (B-AMC) and P34 system, for air traffic management (ATM) to achieve a ground–air data link connection fusion on the road [10,11,12]. Xihe, a seamless combination indoor and outdoor positioning system, solves the problem of the positioning and navigation access methods used by the end-users in the Asia-Pacific [13] region. Similarly, the Hongyan constellation plan was proposed by China and provides both broadband communication and space-based augmentation system (SBAs) services to end-users all over the world [14,15].

With the large-scale application of 5G communication technology and the rapid development of multi-function micro-satellites or pico-satellites, signal design carried out under a multi-source signal system aiming at achieving “one world, one network” has become a “conduction fusion” (NAV/COM), which is the key link needing to be solved urgently. Prof. Zhongliang Deng and his team from Beijing University proposed a TC-OFDM scheme integrated signal system for NavCom, which forms high-precision synchronization between ground base stations and satellites. After actual measurement and evaluation have been carried out, the accuracy of the signal can reach the meter level [16,17,18]. Prof. Xiaoli Liu and her team at Wuhan University proposed a signal based on MSK and OFDM modulation and evaluated these two signals through navigation attributes and tracking performance. They believed that these two signals have better performance than other signals studied in the paper [19]. Ruidan Luo et al. proposed a navigation modulation method based on a multi-carrier composite carrier. Compared with the existing modulation signal, it has outstanding performance in terms of tracking accuracy, anti-interference, and spectrum utilization and can even be used as a signal for a ground-based augmentation system [20,21,22].

Regarding the existing GNSS navigation frequency bands, the upper L-band (1559~1610 MHz), the lower L-band (1164~1300 MHz), and the S-band (2483.5~2500 MHz) are allocated to the GNSS system by the International Telecommunication Union (ITU). At present, most GNSS systems use an L-band frequency. Although the bandwidths of the upper L-band and lower L-band are much larger than those of the S-band, the rapidly stacking signal of the GNSS system causes L-band frequency resources to be quickly squeezed. On the other hand, the signal propagation loss at the S-band frequency is greater than that at L-band, but in terms of rain attenuation, ionospheric delay, antenna size, antenna gain, and other factors, the S-band has more advantages than the L-band [23,24]. In addition, a key change has occurred—that is, software-defined radio (SDR) devices have gradually come to replace firmware and the hardware cost will become negligible, especially for transceivers. In short, for the future GNSS, especially the next generation of BDS, the development of the S-band (2483.5 MHz~2500 MHz) has huge potential resource reserves and great experimental worth.

Research on the S-band signal system design: The Galileo system has been used to conduct research on and evaluate a signal system based on in-orbit validation (IOV) in an S-band trial operation, where a series of candidate signals, including BOCs (5, 2) and BPSK (1), were evaluated with the modulation of a binary offset carrier with sine and binary phase-shift keying, respectively [25,26,27]. Rui Xue and Prof. Yanbo Sun et al. proposed the use of continuous phase modulation (CPM) to modulate S-band signals and verified that these have a better tracking accuracy, anti-interference properties, and multipath cancellation performance than other candidate modulated signals [28,29,30]. Lei Wang et al. proposed the use of a minimum-shift keying (MSK) modulation method for radio determination satellite service (RDSS) spread-spectrum signals named SSMSK and found that SSMSK had a better performance in acquisition, tracking, and user capacity through a performance evaluation [31].

This article is organized as follows. Section 1 reviews the main nodes of the current augmentation system, satellite navigation system, and NavCom signal scheme, respectively, and introduces the target frequency band of the signal scheme proposed in this article and previous research results. Section 2 lists the in-band and adjacent-band satellite-based system and ground-based system services at 2483.5 to 2500 MHz and describes and analyzes the main systems in-band. Section 3 proposes the signal model and generation model and discusses the basic properties of the proposed signal. Section 4 compares the proposed scheme with the in-band spread spectrum signals of existing systems and a candidate signal scheme from the perspective of the performance method, with indicators such as normalized power spectral density (NPSD), Gabor bandwidth, code tracking error, spectral sensitivity coefficient (SSC), code tracking spectral sensitivity coefficient (CT-SSC), multipath error envelope, link budget tradeoff, and bit error rate (BER), as well as discussing the result of the assessment. Finally, Section 5 concludes the paper and discusses the outlook.

## 2. Related Surveys

The aspect that has the primary impact on the S-band signal design is the mutual compatibility issue, which comes from existing (or planned) systems in-band and in adjacent bands.

From a systematic perspective, there are a large number of signal sources distributed across the dimensions of space, air, ground, and sea. GEO/MEO/LEO constellations in space, LTE 4G/5G base stations, navigation base-stations on the ground, and Wi-Max and Wi-Fi transmitters on vessels, all emit radio waves all the time. Additionally, a large number of receivers are distributed across space–air–ground–sea networks; these detect, acquire, and track radio waves to provide PNTRC services for various users, such as aircraft, vehicles, and vessels, etc. The service scenario diagram is shown in Figure 1, and the main systems are listed in Table 1 [32,33,34,35].

NavCom signals on LEOs, as a subset of the PNTRC source, also face many difficulties, including:Spectrum resources are extremely limited, meaning that it is impossible to give exclusive frequency bands to each system;Chaotically propagating signals cause a sharp decline in system performance due to mutual interference, meaning that they cannot be reused.

For the above-mentioned bottlenecks, it is necessary to bear in mind the following issues in signal design:Spectrum reuse or spectrum efficiency—that is, the work performed per hertz;Planning and design between signals of the same frequency, especially the verification interference/anti-interference and mutual compatibility of planned signals in the same band;In addition to the traditional time–frequency features, mining novel dimensional features is also extremely important for the extraction and distinguishing of ubiquitous signal designs for PNTRC.

### 2.1. Satellite-Based Systems

Satellite-based augmentation systems broadcast various error terms and other calibration information to end-users via GEO/MEO/LEO constellations to improve users’ experience in positioning and timing services. It may be a redundant satellite system independent of GNSS but can also be the communication payload carried on the GNSS satellite.

However, the signal interference encountered on the space link between the satellite and user will be the main challenge facing SBAS services, especially LEO SBAS. Figure 2 shows the normalized power spectral density function of the main electronic systems used for navigation, communication, and other services in the target band. However, it seems that the challenges from actual interferences and threats are not limited to RDSS, IRNSS, and Globalstar, etc., but also affect many undisclosed LEO micro-satellites and high-altitude aircrafts.

In Figure 2, one can see that the signal normalized PSD of BPSK (4) marked by the dark red solid line has a wide main lobe and side lobes and a large value for the peak sidelobe ratio; all these are beneficial for the receiver’s power spectrum detection, tracking, and positioning accuracy performance. The green line marks the BPSK (1) signal with a simple modulation and the main lobe; thus, it is easy for low-cost receivers to detect and track it. The black solid line marks the signal of BOCs (5, 2) and provides a restricted service belonging to the IRNSS. It has the characteristics of the main lobe being split and the width being large, but it is not in the middle area of the target band. For the receiver, the bandwidth determines that its accuracy must be higher than that of the standard positioning service, and the offset of the main lobe determines that more advanced receivers must be used to achieve spectrum detection and tracking. The pink line marks SRC (0.2, 1) and has the characteristics of a certain bandwidth and the same main sidelobe power spectrum. The spectral characteristics can be locked according to the position of the center frequency—that is, the spectral tracking of the receiver can be realized. At the same time, it is different from several other signals in that it has a flat peak duration in its NPSD and is very suitable for data transfer; thus, it can be used as a signal model for NavCom.

From the perspective of the system, RDSS with a frequency range of 2483.5 to 2500 MHz cannot be independently performed by the user for satellite to receiver ranging and position calculation but rather must be coordinated by the external system through the user’s response. According to the regional and frequency allocation regulations in the ITU, RDSS is the main service used in the United States and the secondary service used in Europe, Africa, the Middle East, North Asia, and the Asia-Pacific region. At the same time, ITU-R research supports the use of RDSS/MSS global integration solutions for protecting Globalstar, VICs, and other services [36,37,38]. IRNSS is an area navigation system promoted by ISRO and the ICD. It has been shown that the carrier center frequency of IRNSS is 2492.028 MHz; its signal modulation schemes are BPSK (1) and BOC (5, 2); and it provides SPS and RS services, respectively. The minimum received power of the SPS service is −162.3 dBW [39,40,41]. Globalstar is different from the previous two in that it is a global system that continuously provides mobile voice and data communication services. It consists of 48 LEO satellites. To achieve the frequency reuse of a 16.5 MHz beam, the beam is divided into multiple sub-beams, where the bandwidth of each beam can reach 1.23 MHz. The purpose of this is to effectively avoid the occurrence of frequency aliasing. The signal modulation can be described as SRC (0.2, 1), and the 5th to 13th sub-beams are preferred, as they minimize the probability of interference in other systems in the ISM frequency band [42,43,44].

### 2.2. Ground-Based Systems

Ground-based augmentation systems (GBASs) are defined as the use of a ground reference station network that receives GNSS signals in real-time to calculate the error and broadcast the error correction value to the user terminal through various ground communication networks to realize the error correction of the local position of the user terminal, thereby improving the user’s service experience, with a real-time accuracy up to the centimeter level.

Although GBAS can provide a great performance jump for the user’s equipment, most systems that rely on broadcast currently rely on wireless communication methods. The openness of wireless communication and the characteristics of terrestrial radio frequency points in the target frequency range and accessories make the design of target signals easier to separate, as well as making their ability to resist system interference stronger.

ITU plans to further expand the S-band range for navigation applications, and the compatibility between the target band-related (in-band and adjacent bands) system and RNSS is obviously becoming more and more important to NavCom. At present, systems related to the target band (in-band and adjacent frequency bands) in the terrestrial include Wi-Fi, LTE 4G/5G, Wi-Max, DECT, ZigBee, and BLE.

From the perspective of signal waveform characteristics, Wi-Fi is a near-field radio transmission technology. The peak power of its transmission is regulated by the IEEE protocols. Specifically, the peak power corresponds to IEEE 802.11 a/n, IEEE 802.11 g, and IEEE 802.11 b, being 30, 20, and 10 dBm, respectively. It also has 14 channels at 20 MHz per channel, and the signal modulation is OFDM [45,46,47]. LTE 4G/5G is the current mainstream in mobile communications. It uses beam and power control to achieve high-speed data transmission. The diversity of its modulation methods, the high efficiency of its spectrum utilization, and the use of high frequencies make it more suitable for exclusive working frequencies [48,49,50]. Wi-Max is a technology that provides wireless communication access for long-distance backbone networks. It has the characteristics of a large available frequency band and high transmitting power, but the probability of occurrence is extremely small for interference in S-band navigation signals [51,52,53]. DECT is a wireless access technology with an isolation frequency. It has a high QoS and is suitable for wireless voice communication, and it is considered to have the potential to enable 5G massive machine-type communications (mMTC) by the European Telecommunications Standards Institute (ETSI) [54,55,56]. ZigBee is a low-power, low-data-rate, small-scale wireless access technical specification based on IEEE 802.15.4. It has 16 channels allocated in the 2.4 GHz frequency band, and each channel has a 2 MHz bandwidth. The transmitted power is generally 0–20 dBm. BLE is a small wireless access technology based on IEEE 802.15.1 that is often used for data exchange. For different application scenarios, the maximum allowable transmit power ranges from −3 to 20 dBm [57,58,59].

## 3. Model Design

In this section, the proposed model will be more comprehensively introduced. Specifically, we include not only the signal scheme design but also the signal generation system design. In addition, basic attributes such as time domain and frequency domain will be analyzed.

### 3.1. System Model

The NavCom system model of CE-OFDM-PM is shown in Figure 3. On LEO satellites, the navigation payload is responsible for converting the navigation message into a binary navigation data stream after information processing, then sending it to the phase modulation mapper, while the communication payload is responsible for the data streams of different rates, which are converted into binary data streams via data stream processing and sent to the phase modulation mapper. It should be noted that the data stream can be either a pseudo-random stream or a communication data stream containing information entropy. The former can be used as a signal for a low-orbit satellite navigation system in the strict sense, while the latter can also take into account data communication while performing navigation tasks. In the phase of modulation mapping, the first step is to multiply the navigation data stream and the communication data stream by bit to obtain serial binary data, but the rate of the communication data stream is often times the rate of the navigation data stream. In this case, to solve parallel data, the navigation data must be sampled and expanded to the same length of the navigation and the communication data stream and then multiplied by bit. In the second step, the serial binary data are segmented, converted into multiple parallel binary data, and mapped to the time-varying phase function. Each parallel time-varying phase function data point is used as an input sub-band and modulated with an orthogonal subcarrier; then, the modulated parallel sub-band data are converted into serial data. Here, the baseband signal modulation is completed. The modulated IQ baseband signal is sent to the radio modulation module and the module multiplies and mixes branches I and Q to obtain the modulated signal. After the RF front-end amplification and other types of amplification, the radio frequency antenna of the low-orbit satellite achieves matching. Then, the signal is broadcast to receivers.

However, in the section of the receiver, the NavCom signal is broadcast in the propagation through the RF antenna, the front-end filter, and the amplifier. Then, it is sampled by a downconverter to achieve analog-to-digital conversion and demodulate the RF signal into a digital intermediate frequency signal. After this, signal parameter estimation is carried out to obtain the coarse-grained carrier frequency and data phase information via a locally generated carrier frequency and pseudo-random code phase that can be accurately tracked. After acquisition and tracking are achieved, synchronization is used to obtain the accurate carrier frequency and code phase, which are used to realize the OFDM demodulation of the baseband signal and the inverse mapping of the PM. The parallel data stream is then converted into a serial data stream and the local pseudo-random code is used to separate the navigation message and the communication message.

### 3.2. Signal Model

In the last section, we gave a more detailed description of the signal generation system. Below, we mathematically describe the CE-OFDM-PM signal scheme. The CE-OFDM-PM signal transmission model is shown in Figure 4.

The time domain mathematical model of the bandpass is given by:(1)s(t)=A⋅Re{exp[jϕ(t)]}
where A is the signal amplitude, Re{⋅} is the real operator, and ϕ(t) is the time-varying phase function. This can be expressed:(2)$$\begin{array}{c} \phi \left(t\right)=\theta_{i}+2\pi hC_{N}\sum_{k=1}^{N}I_{i,k}q_{k}\left(t-iT_{c}\right),\\ iT_{c}\leq t\leq \left(i+1\right)T_{c} \end{array}$$

In Equation (2), $\theta_{i}$ is the phase of the i-th symbol with the memory term, h is the phase modulation index, $C_{N}$ is the normalized constant, and $C_{N}=\sqrt{2/N}$. $I_{i,k}$ denotes the data symbol of the i-th symbol, the k-th sub-carrier is $I_{i,k}\in \left\{\pm 1\right\}$, $T_{c}$ denotes the symbol period, and the k-th sub-carrier shape function $\left\{q_{k}\left(t\right)\right\}$ is defined as:(3)$$q_{k}\left(t\right)=\{\begin{array}{c} \cos2k\pi t/T_{c},0\leq t<T_{c}\\ \sin2\pi \left(k-N/2\right)t/T_{c},0\leq t<T_{c}\\ 0 \end{array}$$

Furthermore, the parallel data symbol can be expressed by:(4)$$I_{i,k}=a_{i,k}\cdot d_{i,k}$$
where $a_{i,k}$ is the binary PRN with {0,1} and $d_{i,k}$ is the navigation data with binary {0,1}.

The subcarrier orthogonality condition holds:(5)$$\int_{iT_{c}}^{\left(i+1\right)T_{c}}q_{k_{1}}\left(t-iT_{c}\right)\begin{array}{c} \cdot \end{array}q_{k_{2}}\left(t-iT_{c}\right)dt=\{\begin{array}{c} T_{c}/2,k_{1}=k_{2}\\ 0,k_{1}\neq k_{2} \end{array}$$

Assume that the phases between different symbols are relatively independent—that is, the memory term $\theta_{i}=0$. According to the Maclaurin series, Equation (1) can be expressed as:(6)$$s\left(t\right)=Ae^{j\sigma_{\phi }p\left(t\right)}=A\sum_{n=0}^{\infty }\frac{{\left(j\sigma_{\phi }\right)}^{n}}{n!}p^{n}\left(t\right)$$
where $\sigma_{\phi }^{2}$ denotes the phase signal variance; $\sigma_{\phi }^{2}={\left(2\pi h\right)}^{2}$; and p(t) is the normalized OFDM signal, which is given by:(7)$$p\left(t\right)=C_{N}\sum_{i}\sum_{k=1}^{N}I_{i,k}q_{k}\left(t-iT_{c}\right)$$

### 3.3. PSD and ACF

This subsection will analyze the normalized PSD and autocorrelation function of the proposed scheme and then conduct a preliminary comparison and analysis of the signal scheme proposed in this research with the current leading low-orbit satellite signal system.

The normalized PSD for the CE-OFDM-PM signal can be described as:(8)$$G\left(f\right)=\frac{T_{c}}{2N}\sum_{k=1}^{N}\sin c^{2}\left[\left(f-\frac{k}{2T_{c}}\right)T_{c}\right]+\sin c^{2}\left[\left(f+\frac{k}{2T_{c}}\right)T_{c}\right]$$
where sinc(⋅) denotes the sampling function operator. Figure 5 shows the normalized PSD of the proposed signal in different numbers of sub-carriers *k*.

As can be seen from Figure 5, for different sub-carriers, the characteristics of normalized PSD are as follows:The main lobe splits with the increase in *k*.The normalized power spectral density difference between the logarithmic main lobe and the main side lobe is about 26.5 dBW/Hz.For *k* = 1, the main lobe bandwidth can reach 3.069 MHz, and for *k* ≠ 1 the bandwidth in the main lobe can reach 2.046 MHz, and the side lobe bandwidth can reach 1.023 MHz.

According to the Wiener–Khintchine theorem, the autocorrelation function with the CE-OFDM-PM signal can be expressed as:(9)$$R\left(\tau \right)=\int_{-B_{r}/2}^{B_{r}/2}\frac{T_{c}}{2N}\sum_{k=1}^{N}\sin c^{2}\left[\left(f-\frac{k}{2T_{c}}\right)T_{c}\right]+\sin c^{2}\left[\left(f+\frac{k}{2T_{c}}\right)T_{c}\right]\cdot e^{j2\pi f\tau }df$$
where $B_{r}$ denotes the pre-filtering bandwidth for receivers. The ACF of the proposed signal is shown in Figure 6.

In Figure 6, it can clearly be seen from the waveform above that for the proposed signal, the autocorrelation function has the characteristic of multiple peaks, and for the same value of k, the chip interval of the autocorrelation peak (valley) value is fixed. This will help to lock the characteristics of the autocorrelation peak when waveform transmission distortion occurs in the receiver spreading code tracking stage.

With the increase in *k*, the auto-correlation peak value keeps narrowing, and the period between the peak (valley) values will decrease, causing the receiver to have a higher precision code tracking performance and better multipath resolution.

The navigation signal complex carrier (NSCC) is a signal scheme with advanced features in the field of low-orbit satellites. It was the LEO satellite NavCom signal scheme proposed by Ruidan Luo, Ying Xu et al., Academy of Optoelectronics, Chinese Academy of Sciences. Thus, the proposed signal will be compared with NSCC in normalized PSD, as shown in Figure 7.

It can be seen in Figure 7 that, compared to the sidelobes of the normalized power spectral density, both OFDM (64) and NSCC (19) have an excellent main lobe bandwidth performance, and both have remarkable main-sidelobe ratios. (PSR). Although the sidelobe bandwidth of NSCC (19) is slightly better than that of OFDM (64), the latter is a signal scheme that is widely used in wireless communication, while the OFDM signal scheme has a natural compatibility and interoperability. The authors are relatively more optimistic about the theoretical feasibility and engineering application prospects of the CE-OFDM-PM signal scheme.

## 4. Performance Evaluation

This section briefly introduces the evaluation indicators used; then, the simulation and algorithm are described; and finally, we conduct an in-depth analysis of the simulation results.

### 4.1. Evaluation Criteria

The performance evaluation of GNSS spread-spectrum modulation mainly focuses on an analysis of the performance of the proposed scheme on the accuracy potential indicator, compatibility ability indicator, anti-multipath error ability indicator, and anti-interference ability indicator.

Gabor bandwidth (also called RMS bandwidth) is a core indicator that is commonly used to test the tracking accuracy of receivers. It can be defined as:(10)$$\beta_{RMS}=\sqrt{\int_{-B_{r}/2}^{B_{r}/2}f^{2}G_{s}\left(f\right)df}$$
where *B_r_* is the bandwidth of the receiver RF front-end, *G_s_* (*f*) is the normalized power density spectrum, and *f* is the carrier frequency.

The measurement index of spreading code tracking error can be divided into coherent early minus-late (EML), coherent early minus-late power (EMLP), and Cramer–Rao lower bound (CRLB). The coherent EML spreads the code tracking error in additive white Gaussian noise (AWGN) and is based on the spread spectrum code tracking loop.

As the product of the coherent time and the tracking loop bandwidth approaches zero and the coherent time difference between the early minus late in reference signal is very tiny, CRLB can be defined as:(11)$$\sigma_{CRB}^{2}=\frac{B_{L}}{{\left(2\pi \right)}^{2}\left(C/N_{0}\right)\int_{-B_{r}/2}^{B_{r}/2}f^{2}G\left(f\right)df}$$

In Equation (11), $\sigma_{CRB}$ is the Cramer–Rao lower bound, *B_L_* is the tracking loop bandwidth, $\Delta f_{Gabor}$ is the Gabor bandwidth, *B_r_* is the receiver RF front-end bandwidth, and $C/N_{0}$ is the carrier-to-noise ratio.

The term SSC is used to calculate the amount of noise created when an external signal introduces the receiver together with the input signal in ITU-R M.1831. It is used to measure the interaction and interference between different signals in-band.
(12)$$\chi_{I}^{s}=\int_{-B_{r}/2}^{B_{r}/2}G_{s}\left(f\right)G_{I}\left(f\right)df$$

In Equation (11), *B_r_* is the bandwidth of the receiver RF front-end, *G_s_ (f)* is the normalized PSD with the desired signal, and *G_I_ (f)* is the normalized PSD with the interference signal.

The term CT-SSC is a function used to estimate the influence of interference signals on the tracking performance of the desired signal under a specific spreading code tracking loop discriminator. 

When the correlation distance *d* is relatively small, CT-SSC can be expressed as:(13)$$\begin{array}{c} \kappa \end{array}=\frac{\int_{-B/2}^{B/2}G_{s}\left(f\right)G_{I}\left(f\right){\left(d\pi f\right)}^{2}df}{\int_{-B/2}^{B/2}G_{s}\left(f\right){\left(d\pi f\right)}^{2}df}$$
where *B* is the transmission bandwidth, *G_s_* (*f*) is the normalized power density spectrum of the desired signal, *G_I_* (*f*) is the normalized PSD in the interference signal, and *f* is the carrier frequency.

The mathematical model of the multipath error envelope (MEE) can be expressed as:(14)$$\varepsilon_{\tau }\approx \frac{\pm \int_{-B/2}^{B/2}G\left(f\right)\sin\left(\pi fd\right)\left[\sum_{i=1}^{m}a_{i}\sin\left(2\pi f\tau_{i}\right)\right]df}{2\pi \int_{-B/2}^{B/2}f\cdot G\left(f\right)\sin\left(\pi fd\right)\left[1\pm \sum_{i=1}^{m}a_{i}\cos\left(2\pi f\tau_{i}\right)\right]df}$$

In Formula (14), *B* is the pre-filter bandwidth, *G*(*f*) is the normalized PSD of the signal, *d* is the correlation distance, *α_i_* and *τ_i_* are the amplitude and delay with the *i*-th multipath of the NLOS signal, and *m* is the total number of multipaths. 

In anti-narrowband interference, the terms demodulation anti-jamming narrowband merit factor (Dem and AJNM), anti-matched-spectrum-jamming merit factor (Dem and AJMS), anti-narrowband-jamming merit factors (CT and AJNM), and anti-matched-spectrum-jamming merit factor (CT and AJMS) can be expressed as:(15)$$Q_{DemAJNM}=10\times \log_{10}\left(\frac{1}{R\times \max\left[G_{s}\left(f\right)\right]}\right)[dB]$$
(16)$$Q_{DemAJMS}=10\times \log_{10}\left(\frac{1}{R\times \int_{-B/2}^{B/2}G_{s}^{2}\left(f\right)df}\right)[dB]$$
(17)$$Q_{CTAJNM}=10\times \log_{10}\left(\frac{\int_{-B/2}^{B/2}f^{2}G_{s}\left(f\right)df}{\max\left[f^{2}G_{s}\left(f\right)\right]}\right)[dB]$$
(18)$$Q_{CTAJMS}=10\times \log_{10}\left(\frac{\int_{-B/2}^{B/2}f^{2}G_{s}\left(f\right)df}{\max\left[f^{2}G_{s}^{2}\left(f\right)\right]}\right)[dB]$$

From the perspective of communication payload, the satellite communication link budget is defined as:(19)$${\left(C/T\right)}_{dB}=EIRP-L+{\left(G_{r}/T\right)}_{dB}$$
where *C* denotes the carrier power and can represent the product of the information bit rate transmitted by the carrier *R_b_* and energy of the transmission each bit *E_b_*—that is, $C=R_{b}*E_{b}$. Symbol *T* is the equivalent noise temperature at the input of the receiver. *EIRP* denotes the equivalent isotropic radiation power emitted by the transmitter and *EIRP = G_t_* + *P_t_*, where *G_t_* is the antenna gain and *P_t_* is the transmit power. *L* is the total loss, *L* = *L_0_ + L_s_*, *L_0_* is the propagation loss, and *L_s_* is the system loss. *G_r_* denotes the receiver antenna gain. For the noise power spectral density, *N*_0_ = *K*·*T*—that is:(20)$${\left(C/N_{0}\right)}_{dB}={\left(C/T\right)}_{dB}-{\left(k\right)}_{dB}$$

In Formula (20), *k* denotes the Boltzmann constant and *k =* −228.6 dBW/*K*. Thus, the energy per bit to noise power spectral density ratio (*E_b_*/*N*_0_) can be obtained as below:(21)$$\begin{array}{ll} E_{b}/N_{0} & ={\left(C/T\right)}_{dB}-10\cdot \log\left(R_{b}\right)\\ & ={\left(C/N_{0}\right)}_{dB}-{\left(k\right)}_{dB}-10\cdot \log\left(R_{b}\right)\\ & =EIRP-\left(L_{0}+L_{s}\right)+{\left(G_{r}/T\right)}_{dB}-{\left(k\right)}_{dB}-10\cdot \log\left(R_{b}\right) \end{array}$$

### 4.2. Navigation Performance Analysis

In this subsection, we will compare CE-OFDM-PM (2) with signals from major existing systems such as BPSK (1), BPSK (4), BOCs (5, 2), and SRC (0.2, 1). We will also compare it with different parameters and compare it to the classical modulation signals such as BOCs (5, 2.5) and MSK (10), as proposed by Guoping Jin [60] and Avila-Rodriguez [61], respectively.

Accuracy potential

It can be seen from Figure 8 that when the front-end bandwidth of the receiver increases in the range of 7–10 MHz, the Gabor bandwidth increases sharply, and the cost performance between the positioning performance and the overhead is the highest at this time. When the receiver front-end bandwidth is between 10 and 40 MHz, the Gabor bandwidth is constant, which means that the blind receiver front-end bandwidth is cost-effective for the signal scheme proposed in this article.

Figure 8 shows the Gabor bandwidth of different spread spectrum-modulated signals. The higher the frequency of the power spectrum of the modulated signal, the larger the Gabor bandwidth value will be. When the receiver bandwidth B is below 13.5 MHz, the Gabor bandwidth of the proposed scheme is very small. When the receiver bandwidth B = 15 MHz, the Gabor bandwidth of the proposed scheme is equivalent to the Gabor bandwidth of BPSK (1), and when the receiver bandwidth B = 16.5 MHz, the Gabor bandwidth of the proposed scheme is the same as that of MSK (10) The bandwidth is equivalent, reaching 2.3586 MHz, which is better than that of BPSK (4) but lower than the Gabor bandwidth of BOCs (5, 2) and BOCs (5, 2.5) at 4.29469 MHz.

Figure 9 shows the Gabor bandwidth of CE-OFDM-PM at different modulation orders n. It is not difficult to see that CE-OFDM-PM (2), which is shown by the grey dashed line, has a comparatively high receiver front-end bandwidth at 16.5 MHz. The Gabor bandwidth is better than that of the other schemes, allowing the receiver bandwidth to be fixed when the receiver bandwidth is used to obtain the best positioning performance.

Figure 10 shows the code tracking error of different spread spectrum-modulated signals. It can be seen from the figure that when the *C*/*N*_0_ is 15 dB, the code tracking error (grey dashed line) of the proposed scheme is about 1.2 m and the performance is obviously better than that of BPSK (1) and BPSK (4) and slightly better than that of BOCs (5, 2), BOCs (5, 2.5), and MSK (10). Additionally, the positioning error is inferior to that of SRC (0.2, 1). As the noise ratio increases, the code tracking accuracy will sharply improve; when the CNR is 20 dB·Hz, the code tracking accuracy will reach about 0.85 m, which is significantly better than that of other candidate signals in the S-band.

Figure 11 shows that when the modulation order n is different, the code tracking error of the scheme proposed in this paper is slightly smaller than that of CE-OFDM-PM (1) and significantly smaller than that of CE-OFDM (4). Therefore, it can be found that our proposed signal has a good performance in terms of positioning accuracy.

Compatibility

Figure 12 shows the SSC between the target modulation signal and single interference modulation signal. The smaller the SSC is, the smaller the interference between the two will be and the easier it will be to separate the target signal. It can be seen from Figure 12 that the SSC of the CE-OFDM-PM-modulated signal has significantly smaller advantages of SSC than the other signals. The scheme proposed in this paper can even reach a value of −107.9 dB/Hz for the SSC of MSK (10) and performs better for BPSK (1) and BPSK (4). Both BOCs (5, 2.5) have spectral separation coefficients greater than or equal to 100 dB/Hz; thus, compared with the other listed modulation signal schemes, this scheme is able to extract and separate signals after mixing.

Figure 13 shows the code tracking spectral sensitivity coefficient (CT-SSC) between different spread spectrum modulated signals, which is used to measure the spread spectrum code tracking performance of the interference signal on the target signal under the special code tracking loop discriminator. Normally, the smaller the CT-SSC is, the smaller the influence of the interference signal on the tracking performance of the target signal’s spreading code will be. It can be seen from Figure 13 that when the proposed scheme is used as the target signal, the CT-SSC value of other spread spectrum modulated signals is the smallest value in each row of the matrix. As the target signal, the value of the CT-SSC proposed in this article is also the smallest value in each column of the matrix. Therefore, this shows that the solution proposed in this article has a smaller impact on the tracking performance of the target signal spreading code. Other signals have less influence on it.

Anti-multipath

Figure 14 shows the multipath error envelope of different spread spectrum-modulated signals under the condition of multipath delay within 500 m. It can be clearly seen from the envelope line that the solution proposed in this paper (marked by the grey dashed line) is a multipath error and is significantly smaller than that of several other types of spread spectrum-modulated signals, which means that the proposed scheme has a better anti-multipath performance when there are short-distance multipath delays.

Figure 15 shows the multipath error envelope of different spread spectrum-modulated signals under the condition of multipath delay within 4000 m. From the envelope, the multipath error of the scheme proposed in this paper is slightly better than SRC (0.2,1). However, it is far superior to other spread spectrum-modulated signals, which means that CE-OFDM-PM is the best choice of the spread spectrum signal against the multipath in the case of a large multipath delay.

Figure 16 shows the average multipath error envelope of different spread spectrum-modulated signals under the condition of multipath delay within 4000 m. It can be seen from this that the average multipath error of the scheme proposed in this paper is maintained at different multipath delay conditions. Within 0.1 m, this level is significantly ahead of that of other signal systems.

Figure 17 shows the multipath error envelope of CE-OFDM-PM under different modulation orders n. It can be seen from this that under the condition of a short-distance multipath delay of 300 m, the scheme proposed in this paper compared with CE-OFDM-PM (1), CE-OFDM-PM (4), and CE-OFDM-PM (8) has a smaller multipath error envelope.

Figure 18 shows the multipath error envelope of CE-OFDM-PM with different values of n under the long-distance multipath delay condition of 4000 m, as well as the multipath error envelope of the short-distance multipath delay condition of 300 m. The conclusions are similar. Compared with CE-OFDM-PM (1), CE-OFDM-PM (4), and CE-OFDM-PM (8), the scheme proposed in this paper (marked by the grey dashed line) has the advantage of having a long-distance multipath delay. The smaller multipath error envelope means that this scheme has a more significant anti-multipath advantage in the modulation order under the same spread spectrum modulation signal scheme.

Above all, the scheme proposed in this article has an overwhelming advantage over the other listed schemes in terms of its multipath error envelope and average multipath envelope.

Anti-jamming

Figure 19 shows the anti-jamming performance of different spread spectrum-modulated signals when the bandwidth is 16.5 MHz. It is not difficult to see that the tracking and matching spectrum, demodulation and matching spectrum, tracking and narrowband, and merit factors of demodulation and narrowband are 73.297, 23.8182, 61.3182, and 12.1418 dB. These results are only inferior to the other spreading-modulated signals in terms of the tracking and narrowband merit factor and are excellent for the other three indicators. Among the other spreading-modulated signals listed, especially in terms of the demodulation and matching spectrum merit factor and the demodulation and narrowband merit factor, our scheme is significantly ahead of other spread spectrum-modulated signals.

### 4.3. Communication Performance Analysis

As a verification indicator for the NavCom signal, the communication link budget between the LEOs and the terminal is extremely critical and the required parameter settings are shown in Table 2.

According to the parameters shown in the above list, assuming that the propagation channel model is a free space model, the transmission loss can be estimated as follows:$$\begin{array}{ll} L_{0} & =32.45+20\cdot \log\left(f\right)+20\cdot \log\left(D\right)\\ & =32.45+20\cdot \log\left(2491.75\right)+20\cdot \log\left[\left(500\sim 1000\right)\times 2.2\right]\\ & =32.45+67.93+20\cdot \log\left[\left(500\sim 1000\right)\times 2.2\right]\\ & =100.38+{\left(60.83\sim 66.85\right)}_{dB}\\ & ={\left(161.21∼167.23\right)}_{dB} \end{array}$$

For the uplink from the ground terminal transmitter to the LEO receiver, the bit signal-to-noise ratio can be estimated as:$$\begin{array}{l} \;E_{b}/N_{0}=EIRP-\left(L_{0}+L_{s}\right)+{\left(G_{r}/T\right)}_{dB}-{\left(k\right)}_{dB}-10\cdot \log\left(R_{b}\right)\\ \begin{array}{ccc} & & = \end{array}\left(11+3\right)-L_{0}-1-6.5+228.6-10\cdot \log\left(64,000\right)\\ \begin{array}{ccc} & & =13 \end{array}-L_{0}-6.5-48.06+228.6\\ \begin{array}{ccc} & & =187.04 \end{array}-L_{0}\\ \begin{array}{ccc} & & =187.04 \end{array}-{\left[161.21\sim 167.23\right]}_{dB}\\ \begin{array}{ccc} & & =\left[19.81\sim 25.83\right]{}_{dB} \end{array} \end{array}$$

Using Figure 20 as an input, an LEO satellite with an orbit height of 600 km and an orbital inclination of 45° was constructed through simulation verification by STK 11.6, named WuhanSat001. The ground station, named WHUT-FIXED, was set as a longitude of 114.367022° E, the latitude was 30.544342° N, the altitude was 27.1 m, and the transmitter and receiver were established on the satellite and ground station, respectively. The simulation was performed as shown in Figure 20 and Figure 21.

In the verification of the simulation under ideal conditions, only one satellite and one ground station were set up. Thus, the transmitter and receiver realized transmission within eight timeslots covered by the communication; the signal coverage of transceivers between ground stations is shown above. Since the indicators in the simulation results are very complex, it is inconvenient to display all of them in this paper. A few critical indicators in the communication link tradeoff are displayed in Figure 22 and Figure 23.

From Figure 22, it is obvious that, under the simulation conditions, the BER is 1 × 10^−30^ in the uplink, which is far better than the threshold of 1 × 10^−6^. Additionally, *E_b_*/*N*_0_ has a redundancy of 18.4299~29.2671 dB and *C*/*N* has a redundancy of 3.4814~14.3186 dB.

In Figure 23, it is obvious that under the worst conditions, the BER can reach 9.47 × 10^−11^ in the downlink, significantly surpassing than the threshold of 1 × 10^−6^, while *E_b_*/*N*_0_ has a redundancy of 13.072~23.8268 dB and *C*/*N* has a redundancy of 7.0514~17.8062 dB.

From the link communication budget shown below:Our proposed signal scheme is capable of meeting the communication conditions between LEOs and terrestrial locations.The low power of the user terminal and the link bandwidth tradeoff settings of the communication signal design can provide technical support for the miniaturization of the transceiver terminal and multi-user cluster.The scheme has the potential for expansion towards multi-orbit and multi-LEO satellite networking.

## 5. Conclusions and Outlook

Based on the S-band LEO constellation, this article designs a signal scheme suitable for NavCom. It uses CE-OFDM-PM (2) for modulation. At the same time, the signal generation system and process are described in this paper. After simulation experiments, via comparing different signal systems, we found that it has excellent accuracy potential, better compatibility, excellent anti-multipath ability, and outstanding anti-narrowband interference performance. Additionally, it has impressive communication link tradeoff indicators, such as BER, *C*/*N*, and *E^b^*/*N*_0_ etc. From a theoretical viewpoint, it provides a brand-new perspective for the integration of the NavCom of LEOs.

At the same time, the flexibility of CE-OFDM-PM (2) in the signal system also has the potential to achieve interconnection and intercommunication with many wireless terminals. It has significant signal application and engineering application prospects for the larger-scale integration of NavCom and PNTs in the future.

The theoretical feasibility and simulation accuracy of the scheme proposed in this study will provide a foundation for subsequent work on LEO satellite networking, verification work for different types of carriers, and performance testing at different frequencies.

In the future, we will focus our work on the adaptability of spread spectrum-modulated signals in multi-orbit constellations. This will be a core step in the process of CE-OFDM-PM from design to engineering and implementation. On the one hand, we will consider the Doppler effect caused by the relative motion between the receiver and the transmitter. On the other hand, we will also combine various vehicles for use in verification. This will be the main work of our team in the future. Our work will also provide a foundation for a comprehensive roadmap to be made for the PNTRC scheme.

## Figures and Tables

**Figure 1 sensors-21-08235-f001:**
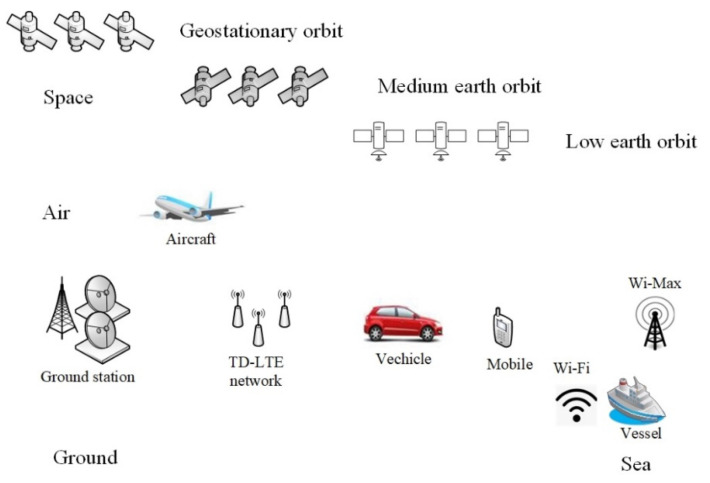
PNTRC service scenario diagram covering space–air–ground–sea integrated networks for the future. Integrated sensing and communication (ISAR) sources such as satellites, air balloons, unmanned aerial vehicles, ground stations, and mobile communication base stations are included, as well as Wi-Fi and Wi-Max transmitters, etc. These are distributed across different dimensions, use different mechanisms, and provide PNTRC services to various users such as aircrafts, vehicles, vessels, and individual mobiles.

**Figure 2 sensors-21-08235-f002:**
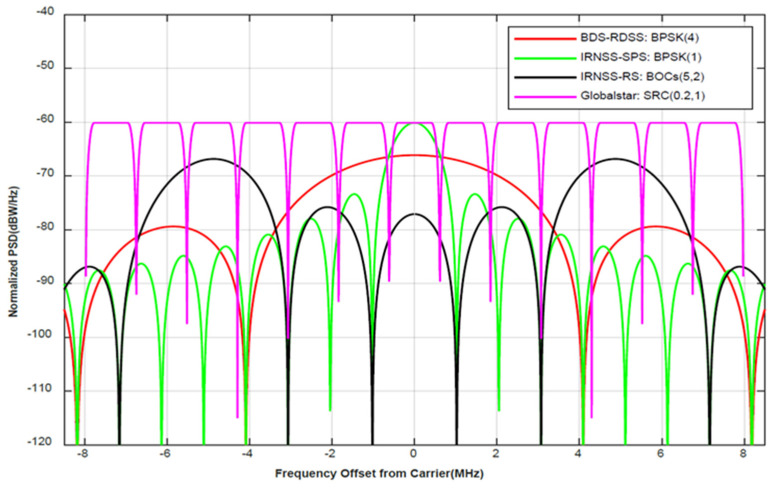
Comparison of normalized power spectral density function of the main satellite–based electronic signals in the S–band. The main peak of the green solid lines’ normalized PSD function is in the center of the receiver, with unnecessary frequency modulation seen for the receiver; the main peak of the black solid lines is split and located on both sides to avoid the aliasing of the main peak and ensure the clarity of the main peak of the spectrum; the main lobe bandwidth of the red solid lines is significantly larger than that of the others, and it has a better code tracking accuracy; the pink solid line is divided into 13 sub–bands with equal bandwidths and pulse-like modulation and is suitable for carrying out the transmission of information.

**Figure 3 sensors-21-08235-f003:**
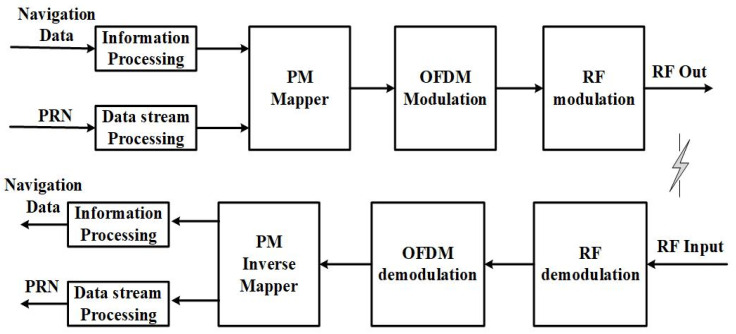
NavCom system model with CE–OFDM–PM signal. Navigation data and communication data (or pseudo-random noise) are sent to the phase mapper together via data processing, then the phase of the symbol is generated, OFDM modulation is performed and moved to the target carrier band via radio frequency modulation and transmission, and the receiving end undergoes demodulation and inverse mapping, resulting in the navigation and communication data, respectively, being separated in the final product.

**Figure 4 sensors-21-08235-f004:**
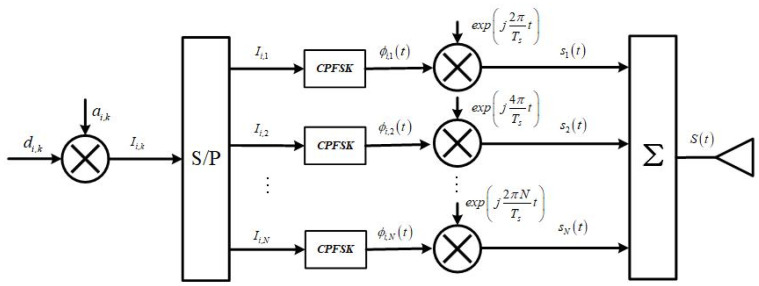
Inner structure of a model of a CE-OFDM-PM signal generator. The navigation data *d_i_* and *_k_* and communication data *a_i_* and *_k_* are operated bitwise. Then, *I_i_* and *_k_* undergo serial-to-parallel conversion, and CPFSK modulation is performed on each parallel bit to obtain the unique phase of the binary data. After Fourier transform is carried out, the sum of each parallel signal is used to find the final transmitted time domain signal.

**Figure 5 sensors-21-08235-f005:**
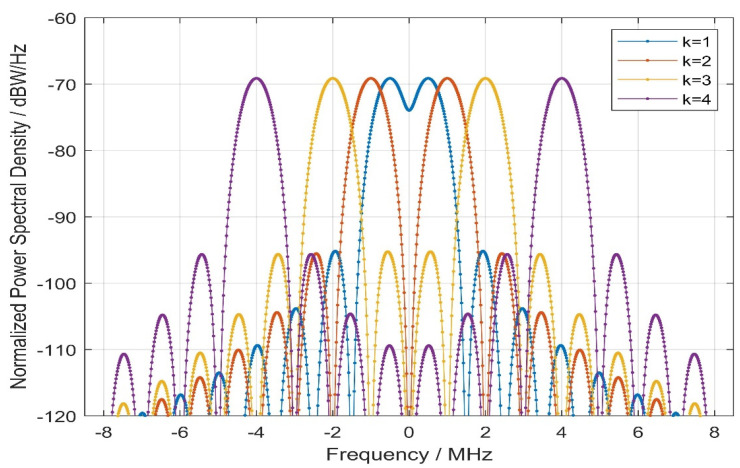
Comparison of the normalized power spectral density with CE–OFDM–PM under the conditions of *k* = 1, 2, 3, 4. With the increase in parameter k, the main peak of the normalized PSD of the CE–OFDM–PM spread spectrum modulation will be split from the center. This flexible feature is similar to the BOC spread spectrum modulation, and the former has a better main-to-side lobe peak ratio than the latter.

**Figure 6 sensors-21-08235-f006:**
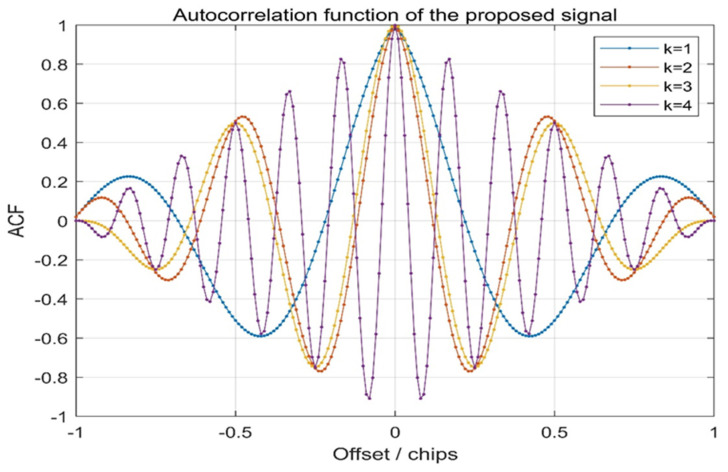
Comparison of autocorrelation function with CE–OFDM–PM under the conditions of *k* = 1, 2, 3, 4. For the increase in k, the peaks (valley) are increased in each period, but for a fixed k, the autocorrelation peaks (valleys) are fixed, meaning that this is suitable in the receiver spreading code tracking stage.

**Figure 7 sensors-21-08235-f007:**
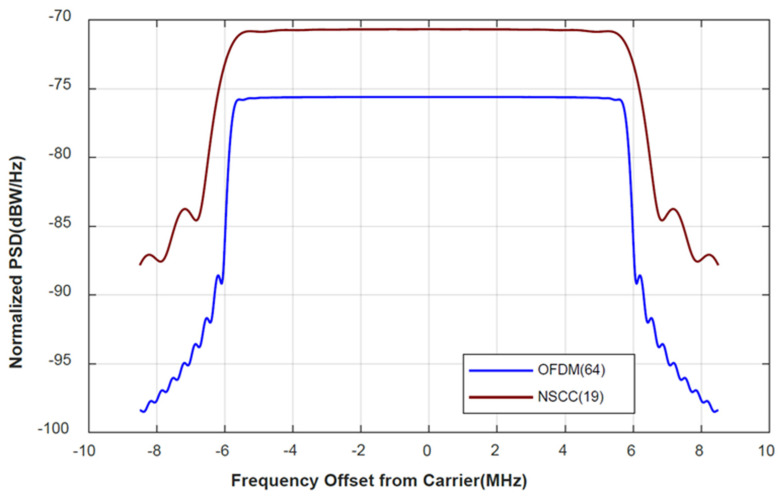
CE–OFDM–PM (64) is compared with NSCC (19) in terms of normalized power spectral density. In the normalized PSD, the scarlet solid line is significantly larger than the blue, and the former supports the multiplexing of more users in the same bandwidth and suppresses the power generated by other signals at the same band. This is in line with typical green radio techniques and helps to improve the energy efficiency.

**Figure 8 sensors-21-08235-f008:**
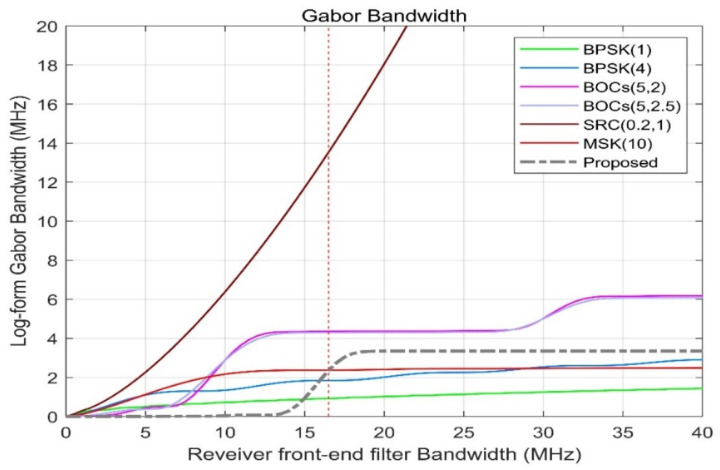
Proposed signals’ Gabor bandwidths compared with BPSK (1), BPSK (4), BOCs (5, 2), BOCs (5, 2.5), SRC (0.2, 1), and MSK (10). For bandwidths of 16.5 MHz at the front end of the receiver, the gray dashed line has a Gabor bandwidth that can exceed the green solid line and the blue, while the performance is shown by the dark red.

**Figure 9 sensors-21-08235-f009:**
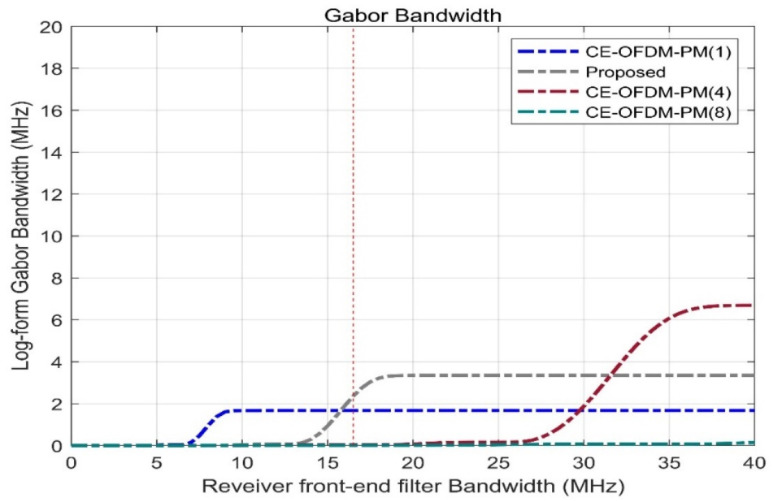
Comparison of the Gabor bandwidth under the conditions of *k* = 1, 2, 4, 8. For the 16.5 MHz receiver front-end bandwidth, the gray dashed line corresponding to *k* = 2 has the best Gabor bandwidth, which means that broadband receivers can have better positioning and resolution capabilities, consistent with NavCom.

**Figure 10 sensors-21-08235-f010:**
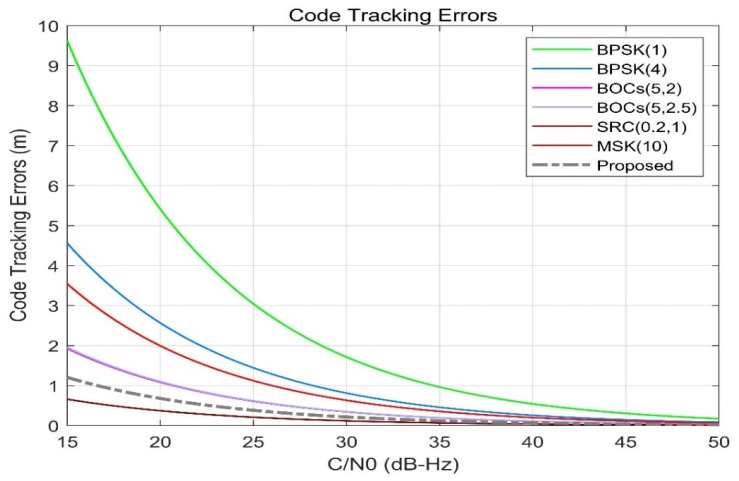
Proposed signals’ code tracking errors compared with BPSK (1), BPSK (4), BOCs (5, 2), BOCs (5, 2.5), SRC (0.2, 1), and MSK (10). For a carrier-to-noise ratio of 15 dB-Hz, the code tracking error indicated by the gray dashed line is about 1.2 m, which is much better than others, except for the dark red. This shows that CE-OFDM-PM (2) has outstanding potential for use in code tracking.

**Figure 11 sensors-21-08235-f011:**
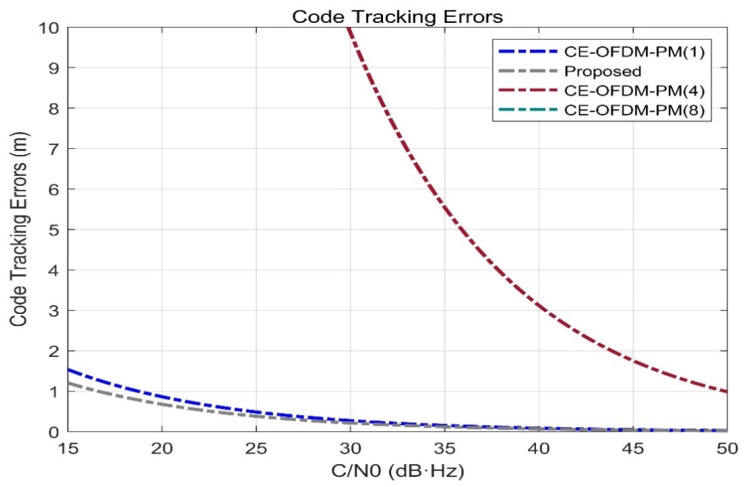
Comparison of code tracking errors with CE-OFDM-PM under the conditions *k*= 1, 2, 4, 8. It can be found that the gray dashed line has more advantages in terms of code tracking error.

**Figure 12 sensors-21-08235-f012:**
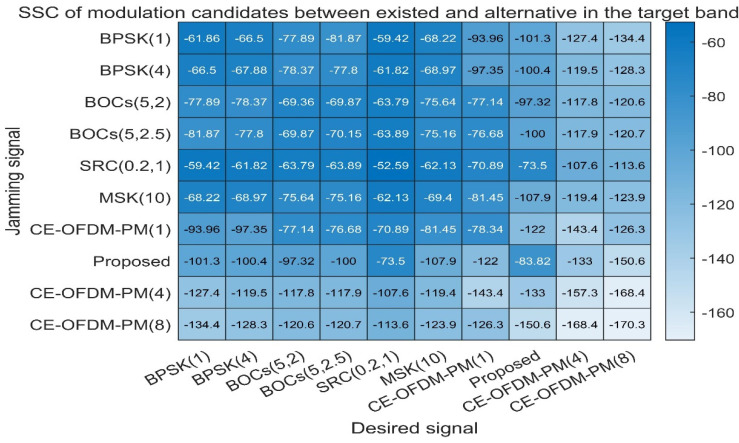
Proposed signals’ spectral separation coefficients compared with BPSK (1), BPSK (4), BOCs (5, 2), BOCs (5, 2.5), SRC (0.2,1), MSK (10), CE–OFDM–PM (1), CE–OFDM–PM (4), and CE–OFDM–PM (8). Obviously, the SSC of our proposed scheme and the above-listed modulation is significantly smaller than that of other spread spectrum signals. Except for CE–OFDM–PM (4) and CE–OFDM–PM (8), which shows that our proposed scheme can easily be compatible with service signals of the same frequency band, while their own compatibility is also remarkable.

**Figure 13 sensors-21-08235-f013:**
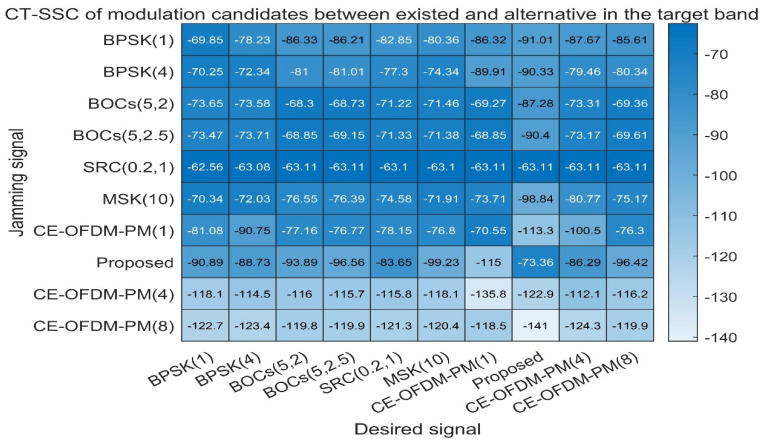
Proposed signals’ code tracking spectral sensitivity coefficients compared with BPSK (1), BPSK (4), BOCs (5, 2), BOCs (5, 2.5), SRC (0.2, 1), MSK (10), CE–OFDM–PM (1), CE–OFDM–PM (4), and CE–OFDM–PM (8). From the perspective of the row or column, except for CE–OFDM–PM (4) and CE–OFDM–PM (8), regardless of whether our proposed signal is a jammer or desired signal, its interference with other signals or the influence of other jammers is very small. Additionally, this indicates that the proposed signal has an outstanding compatibility in the code tracking process.

**Figure 14 sensors-21-08235-f014:**
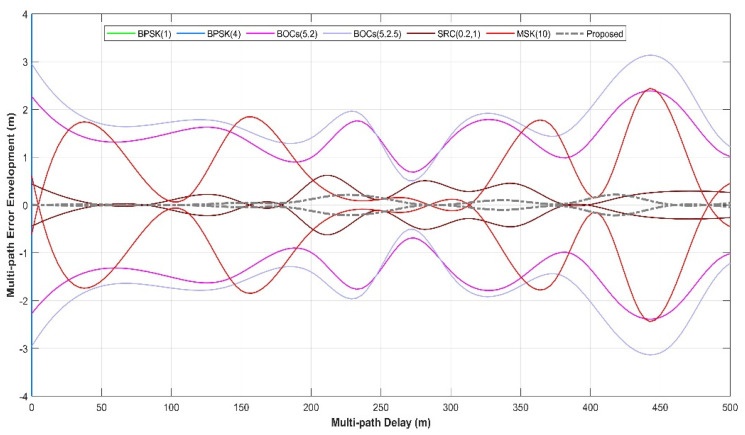
Proposed signals’ multipath error envelope compared with BPSK (1), BPSK (4), BOCs (5, 2), BOCs (5, 2.5), SRC (0.2, 1), and MSK (10) within 500 m. The grey dashed line shows that the multipath error envelope within 500 m of the multipath delay is much smaller than that of other lines. This means that the proposed spread spectrum signal itself has very remarkable multipath suppression capabilities.

**Figure 15 sensors-21-08235-f015:**
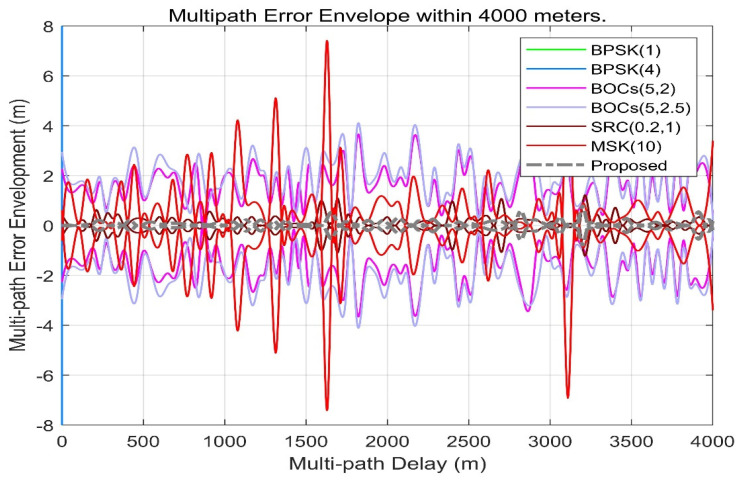
Proposed signals’ multipath error envelope compared with BPSK (1), BPSK (4), BOCs (5, 2), BOCs (5, 2.5), SRC (0.2, 1), and MSK (10) within 4000 m. Obviously, the grey dash line also has a smaller multipath error envelope over long distances. This means that the proposed signal also has a better adaptability in scenarios on more reflective surfaces or longer reflection distances.

**Figure 16 sensors-21-08235-f016:**
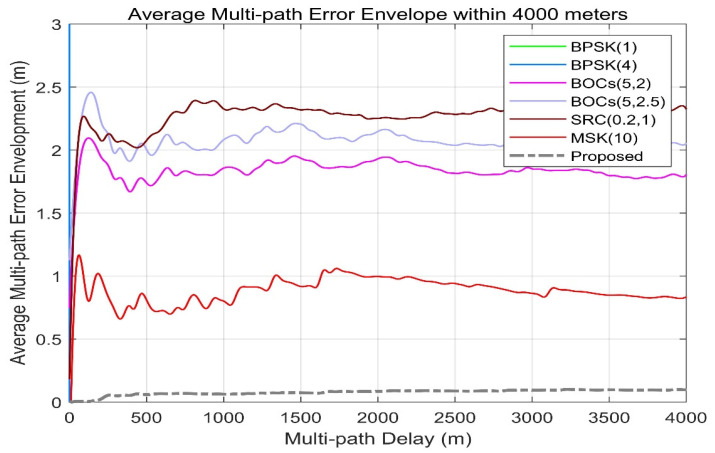
Proposed signals’ average multipath error envelope compared with BPSK (1), BPSK (4), BOCs (5, 2), BOCs (5, 2.5), SRC (0.2, 1), and MSK (10) within 4000 m. In terms of the statistical indicators of the multipath envelope, the grey dashed line maintains an average value of between 0.08 and 0.11 m while the multipath delay is within 4000 m.

**Figure 17 sensors-21-08235-f017:**
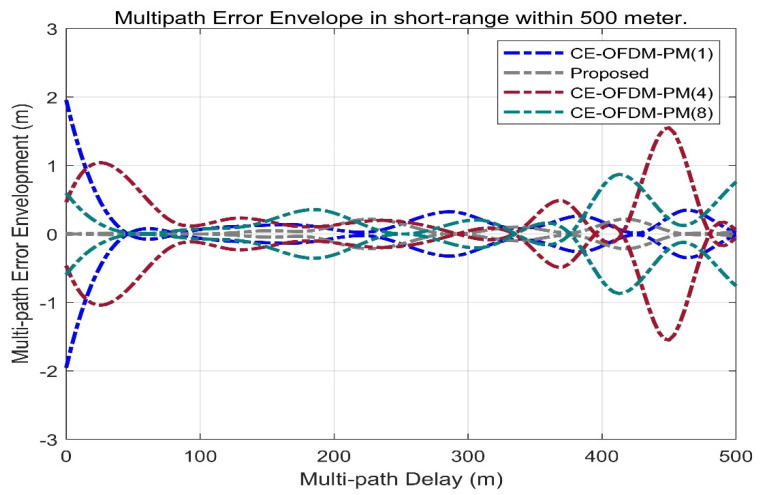
Comparison of multipath error envelope with CE–OFDM–PM under the conditions of *k* = 1, 2, 4, 8 within 500 m. The proposed signal marked by the gray dashed line has a smaller multipath error envelope than the signal of the CE–OFDM–PM signal family within 500 m.

**Figure 18 sensors-21-08235-f018:**
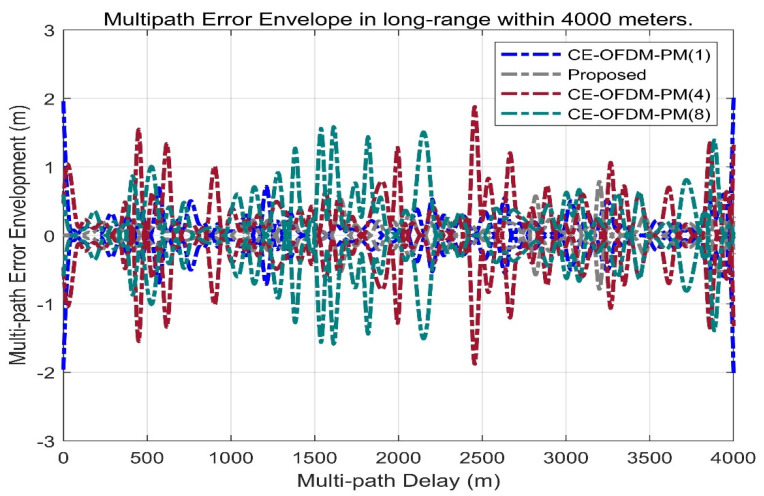
Comparison of multipath error envelope with CE–OFDM–PM under the conditions of *k* = 1, 2, 4, 8 within 4000 m. The proposed signal also has a smaller multipath error envelope than the signal of the CE–OFDM–PM family within 4000 m.

**Figure 19 sensors-21-08235-f019:**
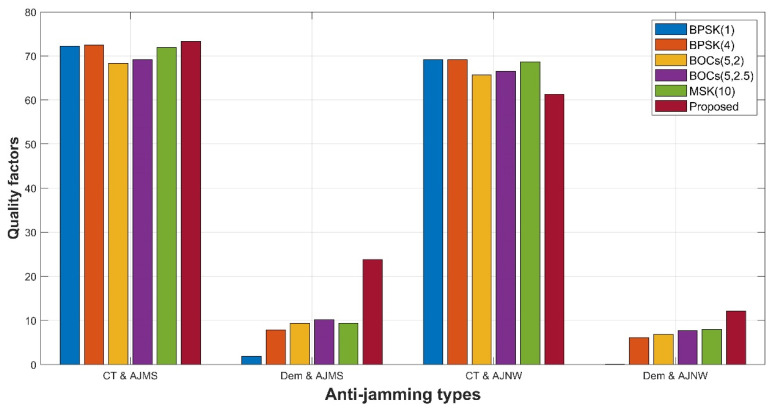
Histogram of anti-jamming level among proposed signal, BPSK (1), BPSK (4), BOCs (5, 2), BOCs (5, 2.5), and MSK (10). The dark red bar shows significantly stronger anti-jamming for the matching spectrum (Dem and AJMS) merit factor and anti-narrowband jamming (Dem and AJNW) merit factor in the demodulation process than other bars, and the code tracking is resistant to the matching spectrum interference (CT and AJMS) merit factor, which is slightly stronger than the others and significantly lower than the others in the code tracking anti-narrowband jammer (CT and AJNW). Our proposed signal has better comprehensive anti-interference characteristics.

**Figure 20 sensors-21-08235-f020:**
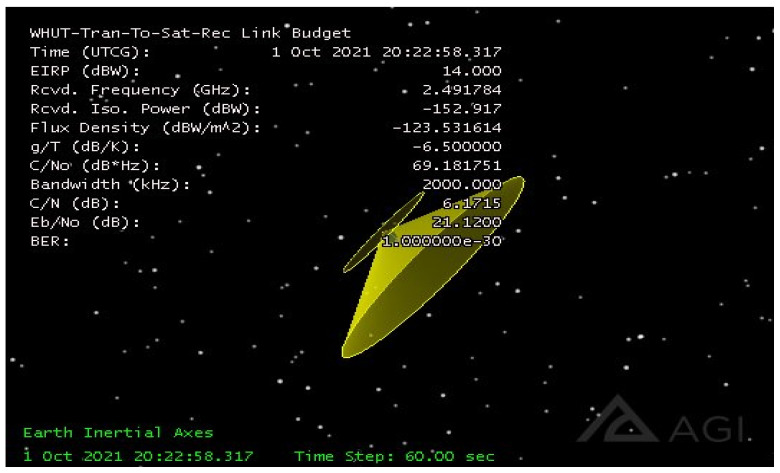
Three-dimensional diagram of the parameter settings. Via STK 11.6 simulation for the above-mentioned parameters, the signal transmitted by the ground station WHUT-FIXED shows that the standard received power on the satellite is −152.917 dBW, while the carrier (signal) power/noise spectral density is about 69.18 dB·Hz. The carrier/noise ratio is about 6 dB, the bit signal-to-noise ratio is 21.12 dB, and the bit error rate is maintained at an ultra-low level of 1 ×10^−30^. This means that the uplink tradeoff performed remarkably in all the above parameters.

**Figure 21 sensors-21-08235-f021:**
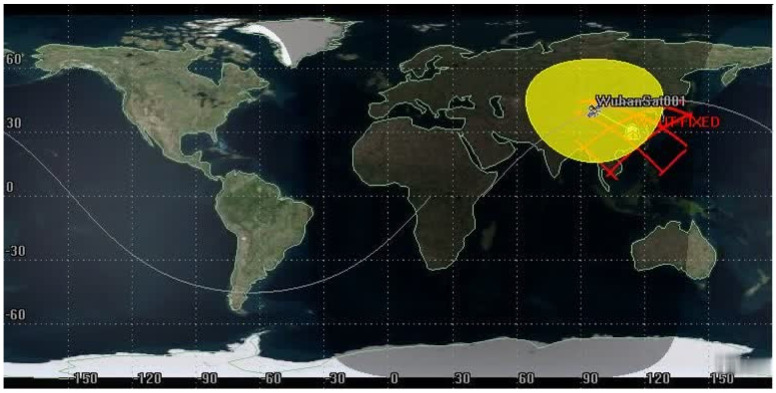
Two–dimensional diagram of the parameter settings. The red solid line represents the orbital area covered by the ground fixed transmitter in the set conditions, while the yellow area denotes the satellite transmitter coverage for the Earth’s surface.

**Figure 22 sensors-21-08235-f022:**
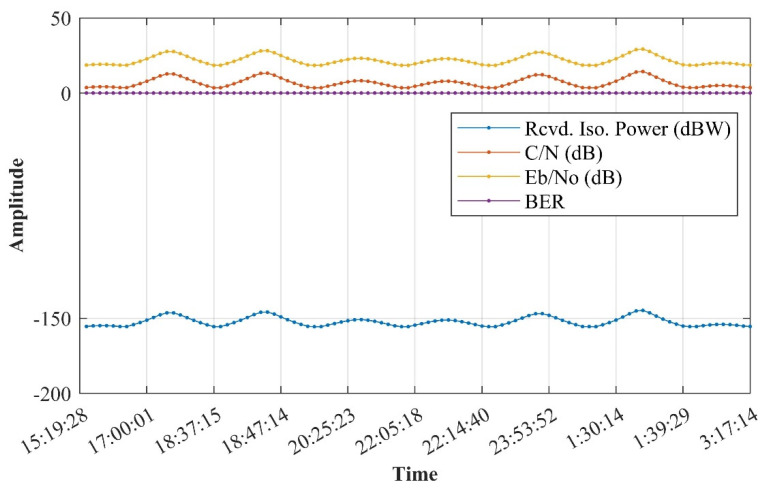
Uplink parameters via STK. With the relative movement between the ground station and the satellite, sampling is performed every 1 min, and the 1–day period is divided into 8 coverage slots. The power received via the uplink, carrier-to-noise ratio, and bit signal-to-noise ratio (Eb/N0) were −155.607 to −144.77 dB, 3.4814 to 14.3186 dB, and 18.4299 to 29.2671 dB, respectively, and the bit error rate (BER) was 1 × 10^−^^30^. These indicators show that the sensitivity of the satellite receiver is equivalent to that of the existing GNSS ordinary receiver, the carrier-to-noise ratio tolerance is very high, and the bit error rate can be kept extremely low. In terms of the communication field, our proposal has significant worth.

**Figure 23 sensors-21-08235-f023:**
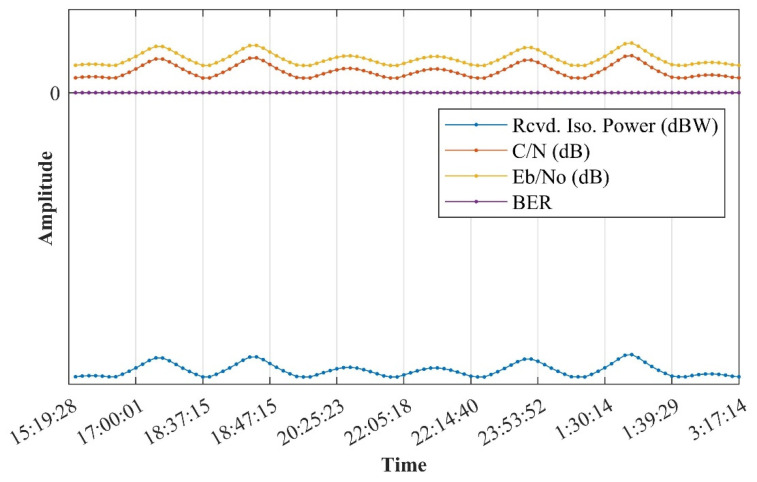
Downlink parameters via STK. According to the above simulation constraints, the received power (Rcvd. Iso. Power), carrier-to-noise ratio (C/N), and bit-to-noise ratio (Eb/N0) of a single satellite signal received on the ground are −136.507 to −125.752 dBW, 7.0514 to 17.8062 dB, and 13.072 to 23.827 dB, and the bit error rate (BER) is from 1 × 10^−30^ to 9.47 × 10^−11^. The power of the receiver is obviously lower than the power level of the communication receiver. At the same time, the carrier-to-noise ratio tolerance and signal quality are maintained at an extremely high level. Considering that the actual low-orbit satellites will be completely covered by multiple constellations, the rate of bit errors will be further reduced, and a remarkable level will be maintained.

**Table 1 sensors-21-08235-t001:** The main parameters of the S-band in-band and in adjacent bands.

Types	System	Freq-Range (MHz)	Bandwidth (MHz)	Modulation	EIRP (dBW)	PWR (dBW)
Space-based	BDS-RDSS	2483.5~2500	16.5	BPSK (4)		−147.8
	IRNSS	2492.028	16.5	SPS: BPSK (1)	/	−162.3~−157.3
				RS: BOCs (5,2)		
	Globalstar	2483.5~2500	16.5	SRC (0.2, 1)		
Planned signal	Galileo Planned			Chirped BPSK (1)	−171.5	
				CBOC (6, 1, 1/11)	−162.3	
	BDS-RNSS Candidate			OFDM		
				CPM (8)		
Ground-based	TD-LTE band 41	2496~2690	20	QPSK/OFDM	−140~−135 (95%)	
	TD-LTE band 53	2483.5~2495	11.5	QPSK/OFDM	−83 dBW/ 100 MHz	−57 dBW/ 100 MHz
	WiMAX	2496~2690	1.75~20	OFDM		
	Wi-Fi Channel 14	2473~2495	22	DSSS/ OFDM	−12 dBW	
	FS	2450~2690		MSK/QPSK		

**Table 2 sensors-21-08235-t002:** List of parameter settings.

Indicator	Symbols	Values
Carrier frequency range	*f*	2483.5~2500 MHz
Carrier center frequency	*f*	2491.75 MHz
Carrier wavelength	*λ*	0.12 m
Bandwidth	*B*	16 MHz
Proposed signal	CE-OFDM-PM	
Propagation distance	*D*	500 km~1000 km
Uplink information transmission rate	*R_b_*	64,000 bps
Downlink information transmission rate	*R_b_*	4,096,000 bps
Satellite antenna gain	*G*	20 dBi
Satellite launch power	*P*	13 dBW
Terminal antenna gain	*G*	11 dBi
Terminal launch power	*P*	3 dBW
Systematic loss	*L* _s_	1 dB
Feeder noise coefficient	*ρ*	0.8
Satellite antenna equivalent noise temperature	*T* _α_	290 K
LNA equivalent noise temperature	*T_LNA_*	80 K
Satellite equivalent noise temperature	*T*	442.2 K
Receiver equivalent noise temperature	*T*	250 K
Uplink	*G_r_*/*T*	−6.5 dB/K
Downlink	*G_r_*/*T*	−13 dB/K

## Data Availability

Not applicable.

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
