# Peer review of "A Novel Signal Design and Performance Analysis in NavCom Based on LEO Constellation"

_sensors, 2021, doi:10.3390/s21248235_

Round 1

Reviewer 1 Report

You have squeezed a lot into this paper, but in doing so, it's important to ensure that your readers don't get lost. I got a bit lost because:

- Some abbreviations are used without introduction (e.g., BDS)
- Some of your equations aren't correct, e.g., (9) uses a t instead of a \tau on the right hand side, and (11) purportedly contains a term for the Gabor bandwidth which isn't there. Eq. (12) is referred to as (11). Then just after that, you say "when d is relatively small" - and the only d I can see there us the d from the integral (as in "df"). And then it's expressed as \kappa, which we haven't seen before.
- Figure 1 isn't referred to from anywhere and serves which purpose? Generally, your figures have extremely short captions. Use captions to provide a bit more interpretation of the take-away lessons from the figures.
- Your figures / simulations contain a bewildering mix of modulation schemes: Fig 2 differs from Fig 8, 12, and 13. 19 again has a different set of schemes.
- Fig 20 is referred to as Fig 2 and there's a bit of a disconnect between the screen shot in the figure and the text. Also, if the - presumably orbital - altitude is "27.1meter", wouldn't that be a bit on the low side? What are the red lines in Fig 21 about?

Also:

- Quite a few grammar and spelling errors, and sentences that seem to end in the middle followed by the next sentence.
- Your Acknowledgments section at the bottom contain default text
- Many references are incomplete and lack volume, issue, or page number information etc. Others (15) have serious formatting issues. Formatting generally is inconsistent.

Author Response

Response to Reviewers

Dear Reviewers:

We are very appreciative of your comments on this article. Through careful reading and thinking of the comments, the authors are convinced that the comments can help to increase academic worth to our work. Thus, we make a meticulous response to each comment.

Specific responses to the comments are shown as follow,

Concern # 1: Some abbreviations are used without introduction (e.g., BDS)

Author response: Accepted and revised.

Author action: We updated the manuscript and revised all abbreviations and make sure to give the full names on the first use. Thanks for your reminder.

Concern # 2: Some of your equations aren't correct, e.g., (9) uses a t instead of a \tau on the right-hand side, and (11) purportedly contains a term for the Gabor bandwidth which isn't there. Eq. (12) is referred to as (11). Then just after that, you say "when d is relatively small" - and the only d I can see there is the d from the integral (as in "df"). And then it's expressed as \kappa, which we haven't seen before.

Author response: Accepted and revised.

Author action: We have revised to the reviewers’ concerns. The "\t" on the right-hand side in Eq. (9) is instead "\tau", and the text "Eq. (11)" at the bottom of Eq. (12) is replaced with "Eq. (12)", and the text "...d is relatively small, it can be..." above Eq. (13) is updated to "...the correlation distance d is relatively small, the term CT-SSC can be...". It will help readers understand the content of this article more easily.

Concern # 3: Figure 1 isn't referred to from anywhere and serves which purpose? Generally, your figures have extremely short captions. Use captions to provide a bit more interpretation of the take-away lessons from the figures.

Author response: Accepted, reply and revised.

Author action: To respond to reviewers’ concerns. Figure 1 is a schematic diagram, is used to illustrate the "air-sky-ground-sea" service scenario with PNTC or PNTRC, which is consistent with the text above in Figure 1 and Table 1 below, and is used to intuitively interpret the main systems in the S-band in the service scenario. For the second half of the suggestion, we accept, we provide more explanation in the image caption.

Concern # 4: Your figures / simulations contain a bewildering mix of modulation schemes: Fig 2 differs from Fig 8, 12, and 13. 19 again has a different set of schemes.

Author response: Accepted and revised.

Author action: To response to your concerns, we explained the spread spectrum signals for comparison in section 4.2, and added explanations to the corresponding text in Figure 2, Figure 8, Figure 12, Figure 13 and Figure 19. To enhance the readability of this article.

Concern # 5: Fig 20 is referred to as Fig 2 and there's a bit of a disconnect between the screen shot in the figure and the text. Also, if the - presumably orbital - altitude is "27.1meter", wouldn't that be a bit on the low side? What are the red lines in Fig 21 about?

Author response: Accepted, revised and reply.

Author action: Respond to your concerns, we carefully checked and revised "Figure 2" in the upper text of Figure 20 to "Figure 20". Also, to reply your question, altitude is "27.1 meters" means the height of the ground fixed-transceiver in the simulation, does not refer to the altitude of low-orbit satellites. And the red line in Figure 21 that you are concerned about in the review comments. The red solid line represents the orbital area covered by the ground fixed transmitter in the set conditions, that is, when the LEO satellite runs to this area, it can meet the uplink communication conditions between the ground and satellite. The red solid line represents the orbital area covered by the ground fixed transmitter in the set conditions, that is, when the LEO satellite runs to this area, it can meet the uplink in communication conditions between the ground and satellite.

Concern # 6: Quite a few grammar and spelling errors, and sentences that seem to end in the middle followed by the next sentence.

Author response: Accepted, checked and revised.

Author action: Regarding the grammar and spelling of the article you raised, we first carefully conducted and checked it, and then you can carefully read the language revision and completion to ensure the authenticity and script of the paper.

Concern # 7: Your Acknowledgments section at the bottom contain default text

Author response: Accepted and revised.

Author action: We have removed the default text to ensure the standardization of the manuscript. Thanks for your reminder.

Concern # 8: Many references are incomplete and lack volume, issue, or page number information etc. Others (15) have serious formatting issues. Formatting generally is inconsistent.

Author response: Accepted and revised.

Author action: Respond to your concerns, we have updated all the references, in accordance with the specifications of the template.

Reviewer 2 Report

This is an interesting and useful article that provides a good communication algorithm.

There are some problems below:
1) Can PNTRC be added to the background of the first paragraph? Because remote sensing is also an important satellite, in particular, China has issued many satellites.

2) The bottleneck problem of integrate NavCom of LEO can be further elaborated.

3) Can the test part's adaptability to multi-track be discussed further?

Some minors:
4) Line 24: The article refers to the location of air, sky and earth, should it be unified as ground-air-space. Sky is a less formal term, and it involves space in your figure.

5)Line 31: Does the abbreviation of this sentence correspond to the first letter of the word in capital?

6) The article involves the name of the person, sometimes the last name is first, sometimes the first name is first, and it needs to be unified. E.g. line 63 and line67.

Author Response

Response to Reviewers

Dear Reviewers:

We are admire to your professional comments on this article. Through careful reading and thinking of the comments, the authors are convinced that the comments can help to increase academic worth to our work. Thus, we make a meticulous response to each comment.

Specific responses to the comments are shown as follow.

Reviewer #2

Concern # 1: Can PNTRC be added to the background of the first paragraph? Because remote sensing is also an important satellite, in particular, China has issued many satellites.

Author response: Accepted and revised.

Author action: In the second paragraph of the paper, the corresponding content of PNTRC is added, and pointed out that PNTRC is the fact that ISAC (Integrated Sensing and Communication) in the future B5G/6G. Thanks for your suggestion.

Concern # 2: The bottleneck problem of integrate NavCom of LEO can be further elaborated.

Author response: Accepted and revised.

Author action: In the 2nd section of the article, above the sub-heading 2.1, we added the text to describe the bottleneck in LEO integrated NavCom. Thanks for your suggestion.

Concern # 3: Can the test part's adaptability to multi-track be discussed further?

Author response: Reply and partially revised.

Author action: We have noticed your concern, and we have responded moderately in the conclusions and outlook. Due to the limited space of the paper, the authors believe that no more in-depth discussion will be made in this article. But this issue is also a matter of great concern to the authors. In fact, the current work of our group has covered the issue of satellite multi-orbit adaptability that you are concerned about. On the one hand, this issue is related to the orbital height and density of the satellite, and on the other hand, it is also The Doppler effect caused by the relative motion between the receiver and the transmitter needs to be considered. We will choose to combine this work in future research for publication. Thanks for your suggestions.

Concern # 4: Line 24: The article refers to the location of air, sky and earth, should it be unified as ground-air-space. Sky is a less formal term, and it involves space in your figure.

Author response: Accepted and revised.

Author action: We admire your meticulousness and accept your suggestions. We have carefully checked all the terms involved in the full text. Lines 275 to 282 of the manuscript are revised to "sea-ground-air-space", and the illustrations, table titles and content are also revised to ensure academic norms in the article.

Concern # 5: Line 31: Does the abbreviation of this sentence correspond to the first letter of the word in capital?

Author response: Accepted and revised.

Author action: Thank you for your reminder. We have changed the 31st line of the manuscript "provides" to "offers" to avoid repetition of the first letter of consecutive words.

Concern # 6: The article involves the name of the person, sometimes the last name is first, sometimes the first name is first, and it needs to be unified. E.g. line 63 and line67.

Author response: Accepted, checked and revised.

Author action: We have carefully checked the names of the people in the full text of the manuscript and revised all names to maintain consistency, including the flaws you pointed out. Thanks for your reminder.

Round 2

Reviewer 1 Report

This looks a wee bit better. What I'd still like to see also are figure captions that don't just say what is shown in the figures, but also what can "take home" from them - what am I as the reader meant to learn from the figure? E.g., when you compare different schemes in a figure, say where your proposed scheme does better etc.

There are also some typos / language issues in the modified text.

Author Response

Dear Reviewers:

We are admiring  your professional comments on this article. Through careful reading and thinking of the comments, the authors are convinced that the comments can help to increase academic worth to our work. Thus, we make a meticulous response to each comment.

Specific responses to the comments are shown as follow,

Review#1

Concern # 1: This looks a wee bit better. What I'd still like to see also are figure captions that don't just say what is shown in the figures, but also what can "take home" from them - what am I as the reader meant to learn from the figure? E.g., when you compare different schemes in a figure, say where your proposed scheme does better etc.

Author response: Accepted and revised.

Author action: Thanks for your professional advice. To respond to reviewer concerns, we have briefly explained the pictures in the caption of each illustration in this article and pointed out their meanings to ensure that readers can quickly understand the value and significance of our work.

Concern # 2: There are also some typos/language issues in the modified text.

Author response: Accepted and revised.

Author action: Thank you for your meticulousness check. To respond to reviewer concerns, we have modified and polished the language of the manuscript via the journal English editing.

Reviewer 2 Report

I am happy to find that all my worries have been resolved. Thank you for your hard work.

Author Response

Comment: I am happy to find that all my worries have been resolved. Thank you for your hard work.

Author response: Thanks for all your comments, they will be part of the academic worth and engineering value of this article.